# Methylxanthines Inhibit Primary Amine Oxidase and Monoamine Oxidase Activities of Human Adipose Tissue

**DOI:** 10.3390/medicines7040018

**Published:** 2020-04-02

**Authors:** Wiem Haj Ahmed, Cécile Peiro, Jessica Fontaine, Barry J. Ryan, Gemma K. Kinsella, Jeff O’Sullivan, Jean-Louis Grolleau, Gary T.M. Henehan, Christian Carpéné

**Affiliations:** 1Institute of Metabolic and Cardiovascular Diseases, INSERM, UMR1048, Team 1, 31432 Toulouse, France; hajahmedwiem608@gmail.com (W.H.A.); cecile.peiro@univ-tlse3.fr (C.P.); jessica.fontaine@inserm.fr (J.F.); 2I2MC, University of Toulouse, UMR1048, Paul Sabatier University, 31432 Toulouse, France; 3School of Food Science & Environmental Health, Technological University Dublin, D07 ADY7 Dublin 1, Ireland; barry.ryan@TUDublin.ie (B.J.R.); Gemma.Kinsella@tudublin.ie (G.K.K.); gary.henehan@TUDublin.ie (G.T.M.H.); 4School of Dental Science, Trinity College, D02 F859 Dublin 2, Ireland; josulli@tcd.ie; 5Department of Plastic Surgery, CHU Rangueil, 31059 Toulouse, France; grolleau.jl@chu-toulouse.fr

**Keywords:** adipose tissue, copper-containing amine oxidase, semicarbazide, monoamine oxidase, hydrogen peroxide, DMSO, methylxanthine, theobromine, caffeine

## Abstract

**Background:** Methylxanthines including caffeine and theobromine are widely consumed compounds and were recently shown to interact with bovine copper-containing amine oxidase. To the best of our knowledge, no direct demonstration of any interplay between these phytochemicals and human primary amine oxidase (PrAO) has been reported to date. We took advantage of the coexistence of PrAO and monoamine oxidase (MAO) activities in human subcutaneous adipose tissue (hScAT) to test the interaction between several methylxanthines and these enzymes, which are involved in many key pathophysiological processes. **Methods:** Benzylamine, methylamine, and tyramine were used as substrates for PrAO and MAO in homogenates of subcutaneous adipose depots obtained from overweight women undergoing plastic surgery. Methylxanthines were tested as substrates or inhibitors by fluorimetric determination of hydrogen peroxide, an end-product of amine oxidation. **Results:** Semicarbazide-sensitive PrAO activity was inhibited by theobromine, caffeine, and isobutylmethylxanthine (IBMX) while theophylline, paraxanthine, and 7-methylxanthine had little effect. Theobromine inhibited PrAO activity by 54% at 2.5 mM. Overall, the relationship between methylxanthine structure and the degree of inhibition was similar to that seen with bovine PrAO, although higher concentrations (mM) were required for inhibition. Theobromine also inhibited oxidation of tyramine by MAO, at the limits of its solubility in a DMSO vehicle. At doses higher than 12 % v/v, DMSO impaired MAO activity. MAO was also inhibited by millimolar doses of IBMX, caffeine and by other methylxanthines to a lesser extent. **Conclusions:** This preclinical study extrapolates previous findings with bovine PrAO to human tissues. Given that PrAO is a potential target for anti-inflammatory drugs, it indicates that alongside phosphodiesterase inhibition and adenosine receptor antagonism, PrAO and MAO inhibition could contribute to the health benefits of methylxanthines, especially their anti-inflammatory effects.

## 1. Introduction

Methylxanthines are phytochemicals found at high concentrations in tea, coffee, and chocolate. Among them, theophylline, theobromine, and caffeine are the best known and are often described as mild psychostimulants [1,2,3]. In fact, caffeine is used daily as a somnolytic by millions of consumers, and as a result, methylxanthines are among the most widely consumed natural medicines, alongside polyphenols, alkaloids, and terpenes. A preventive action against neurodegenerative diseases has been attributed to moderate consumption of methylxanthines from food sources or from coffee [4,5].

The mechanisms of action of methylxanthine psychoactive activity are thought to involve phosphodiesterase inhibition and antagonism of adenosine receptors (thereby modulating cAMP and intracellular calcium in brain) [2]. Indeed, prolongation of cAMP action in peripheral tissues has led to theophylline and related methylxanthines being widely prescribed for the treatment of bronchial [6] and coronary spasm [7]. There are also anti-inflammatory actions attributed to methylxanthines [8,9] although the mode of action remains more elusive than for their psychostimulant or dilator activities [8,10,11]. The recent demonstration of an inhibitory effect of methylxanthines on bovine amine oxidase [12] is in agreement with such anti-inflammatory actions. Bovine plasma amine oxidase is a copper-containing amine oxidase, an ortholog to the product of the human *AOC3* gene (Amine Oxidase, Copper-Containing 3) [13,14]. This enzyme, also called semicarbazide-sensitive amine oxidase or vascular adhesion protein-1 (SSAO/VAP-1) is involved in mediating lymphocyte extravasation at sites of inflammation [15]. Pharmacological inhibitors of SSAO/VAP-1 are under development and are currently being considered for potential use as novel anti-inflammatory drugs [16,17].

The hypothesis that natural methylxanthines can inhibit human SSAO/VAP-1 has been proposed [12], but not yet verified. Such inhibition might contribute to the anti-inflammatory effect of methylxanthines. It is interesting to note that several SSAO/VAP-1 inhibitors have exhibited anti-obesity properties in animal models [18], and the inhibitor, semicarbazide, is known to limit body mass gain and adiposity [19]. Since methylxanthines are clearly endowed with anti-obesity properties [20,21], it could be hypothesized that a SSAO/VAP-1 inhibition by methylxanthines could mediate their tendency to reduce body-weight and fat, alongside their well-known antagonism at adenosine receptors and their inhibition of phosphodiesterases reported to occur in the fat-cells of treated obese rodents [22].

Semicarbazide is nowadays considered as an “historical” inhibitor of SSAO/VAP-1 because its potency and selectivity have been surpassed by various recently designed inhibitors [17,23]. In addition, semicarbazide, which has a relatively simple chemical formula (NH_2_–C=O–NH–NH_2_), is classified as toxic since it behaves as an endocrine disruptor [24], and is found as a contaminant in some foods [25]. Consequently, SSAO/VAP-1 was renamed as Primary Amine Oxidase (PrAO) to denote a definition based on its substrate selectivity rather than on its reactivity to a poorly selective inhibitor [26]. Another consequence of semicarbazide toxicity is the current search for natural PrAO inhibitors [18]. In this context, the recent observation that theobromine and related methylxanthines inhibit bovine plasma PrAO [12] has prompted us to explore whether such interaction might be extrapolated to humans. Hence, we have designed dose-dependent studies to test the putative interactions between natural methylxanthines and the human form of PrAO. Thus, the PrAO nomenclature, defined for the copper-containing amine oxidase that does not use FAD as a cofactor (product of the *AOC3* gene) will be used hereafter, although semicarbazide was used as a reference inhibitor throughout the study.

In mammals, PrAO is highly expressed at the surface of inflamed blood vessels, but also in vascular smooth muscles and in adipocytes [15,27]. In light of the relative accessibility of human subcutaneous adipose tissue and its high PrAO level [28], we performed inhibition studies on this human material. Since human adipocytes also express high amounts of monoamine oxidase (MAO), essentially of the MAO-A form [29], we extended our studies to examine the inhibition of native human MAO by methylxanthines. Caffeine and other methylxanthines have previously been reported to inhibit MAO, but mainly in models using purified human recombinant or rodent enzymes [30], and not using the native form as found in human adipose tissue. These studies took into account the fact that micromolar doses of methylxanthines were without any effect on the bovine plasma PrAO, and that detectable inhibition for the bovine enzyme occurred only between 0.1 and 1.0 mM [12]. Consequently, dose-response curves were built from the micromolar to the millimolar range. Methylxanthines are effective at elevated concentrations on other cellular targets (adenosine receptors, phosphodiesterases), and the use of millimolar doses is not a concern on its own. Indeed, a significant concern is that most of the methylxanthines are active on their known targets near their limit of water solubility, which in some cases is very low and needs to be enhanced by the formation of complexes with compounds such as dimethyl sulfoxide (DMSO). It was therefore necessary in the present study to use vehicle(s) that adequately solubilized the xanthine derivatives, but did not alter amine oxidase activity. With this aim, we compared the advantages and disadvantages of DMSO, sodium hydroxide (NaOH), and ethanol (EtOH) as solubilizing agents for methylxanthines.

All the experiments in this study were carried out in homogenates of human subcutaneous adipose tissue (hScAT). This is a unique biological material in that it has high levels of PrAO, while it also contains high amounts of fat, and some sequestration of lipid soluble substances such as methylxanthines is expected. This can make the accurate estimation of kinetic constants difficult and comparison with previous studies using pure recombinant enzyme must be approached with caution. Nonetheless, the following results will show that, among the methylxanthines tested, theobromine and caffeine were significant inhibitors of human PrAO. Furthermore, caffeine, theobromine, and IBMX substantially impaired the MAO activity of human adipose tissue.

## 2. Materials and Methods

### 2.1. Chemicals and Reagents

Benzylamine, methylamine, tyramine, semicarbazide, 3-isobutyl-1-methylxanthine (IBMX), dimethyl sulfoxide (DMSO), hydrogen peroxide, horseradish peroxidase, semicarbazide, all the tested natural methylxanthines and routinely used chemicals were from Sigma-Aldrich-Merck (St. Quentin Fallavier, France), unless otherwise specified. Amplex Red (from FluoProbes) was purchased from InterChim (Montluçon, France).

### 2.2. Subjects and Preparation of Adipose Samples

Samples of human subcutaneous adipose tissue were obtained from a total of 27 overweight women undergoing reconstructive surgery at Rangueil Hospital (Dept. of Plastic Surgery, Toulouse, France), who provided informed consent before inclusion in the study. Their mean age was 47 years, and mean body mass index was 27.5 ± 0.8 kg/m2. Samples of abdominal subcutaneous adipose tissue (ScAT) were obtained from the surgically removed pieces of fat tissue and immediately stored, without any medium, at −80 °C in 35 mL plastic tubes (Nalge Nunc, Rochester, NY, USA). The study was in compliance with the INSERM guidelines and was approved by the French Ministry of Education, Research and Innovation under the document number DC-2008-452. This is approval to harvest elements of the human body obtained from surgery (without additional surgical procedures) with the informed consent of ("absence of opposition to the use of adipose tissue") the donors who are strictly anonymised (sex, age and BMI are the sole biological data taken).

### 2.3. Benzylamine Oxidation by Human Subcutaneous Adipose Tissue (ScAT)

Just before the amine oxidase activity assays, human ScAT samples were thawed, then 1–2 g was homogenized for 30 seconds with a homogenizer (Tissue Master-125, Omni International, Kennesaw, GA, USA) and sonicated for 10 s (Branson sonifier-150, Danbury, CT, USA) at room temperature in 10 volumes of 200 mM phosphate buffer (pH 7.5). Homogenization at room temperature was mandatory to avoid the formation of a fat cake, which at colder temperatures becomes solid and thereby hampers sample distribution and enzymatic reactions.

Homogenates were immediately distributed into 96-well dark microplates (at approximately 50 µg protein/well) and pre-incubated for 10 min without (control) or with 1 mM semicarbazide (abolishing the SSAO activity), or with the tested methylxanthines and their solubilizing vehicles. Incubation was initiated by the addition of 50 µL of substrate at the indicated final concentrations. A fluorimetric method developed for the MAO assay [31] was slightly adapted as previously described [32] to monitor hydrogen peroxide released by the oxidation of 0.05–1 mM benzylamine by ScAT homogenates. Briefly, hydrogen peroxide release was detected by the addition at 50 µL/well of a chromogenic mixture (4 U/mL horseradish peroxidase plus 40 µM of the fluorescent probe Amplex Red), which generates fluorescent resorufin (ex/em: 540/590 nm). Activity of human PrAO was assessed by measuring the fluorescence during a 30-min incubation at 37 °C in 200 mM sodium phosphate buffer (pH 7.5) at a final volume of 200 µL/well. Raw data were collected in a Fluoroskan Ascent microplate reader (Thermo Labsystems, Turku, Finland) and were normalized with reference to a hydrogen peroxide standard curve (from 0.05 to 5 µM H_2_O_2_). As previously reported [33], maximal velocity of native PrAO present in ScAT was reached at 1 mM benylamine. As this was equivalent to an increase of more than four times the baseline (1568 ± 188 vs 337 ± 76 relative units of fluorescence, n = 16, *p* << 0.001), the majority of the inhibition studies were performed at this benzylamine concentration unless otherwise stated.

### 2.4. Methylamine, Tyramine, and Methylxanthine Oxidation by Human ScAT

The same method as above was used for 0.1–1 mM methylamine, another known PrAO substrate [34], and for 0.1–1 mM tyramine, which has been reported to be entirely oxidized by MAO in hSCAT [29]. While the oxidation of 1 mM methylamine generated similar amounts of H_2_O_2_ as 1 mM benzylamine, mostly in a semicarbazide-sensitive manner, tyramine reached only 40% of the benzylamine-induced signal and was inhibited by pargyline.

Furthermore, the fluorimetric amine oxidase assay activity also permitted verification of whether the methyxanthine derivatives on their own were oxidized, or not, by human ScAT. Since none of the methylxanthines tested was capable of increasing the baseline of H_2_O_2_ production by hScAT (not shown), the hypothesis that they behaved as amine oxidase substrates was discarded, and they were further tested as potential inhibitors of PrAO or MAO.

### 2.5. Solubilization of Methylxanthines

Apart from caffeine, which was readily soluble in water, 7-methylxanthine, paraxanthine, and theophylline were dissolved in 50% v/v DMSO, which permitted testing at a higher final concentration of 2.5 mM in the presence of 12.5% v/v DMSO (equivalent to 1850 mM, assuming that pure DMSO is 14.8 M). Theobromine could be adequately dissolved in pure DMSO only and was the lone agent tested at 2.5 mM with DMSO at 3700 mM (25% v/v) final. IBMX was dissolved in 10% ethanol (EtOH), and when present in the assays at 2.5 mM, its vehicle was at a level of 2.5% v/v (equivalent to 428 mM EtOH). Uric acid was solubilized in NaOH 1 M, and its higher dose was tested in the presence of 100 mM NaOH final concentration. For all agents, no reprecipitation was observed in the subsequent dilutions made in water or phosphate buffer. No difference was observed between the extemporaneously made solutions and those thawed after less than two weeks of freezing at −20 °C. The influence of vehicles on both amine oxidase activity and hydrogen peroxide detection is detailed in the Results section. When necessary, a correction was applied to crude data to distinguish the effect of high doses of a methylxanthine from that of its vehicle.

### 2.6. Statistical Analysis

Experimental data are given as mean ± SEM of the indicated number of independent observations and were analyzed with Prism 6 for Mac OS X (from GraphPad software, Inc, San Diego, CA, USA) by a one–way ANOVA followed by Dunnett’s multiple comparisons. Statistical significance was assumed when *p* < 0.05. Non-linear regression analysis was performed with Graph Pad Prism 5.0.

## 3. Results

### 3.1. Influence of Methylxanthines, Amine Oxidase Substrates, and Inhibitors on Fluorimetric Detection of Hydrogen Peroxide

The measurement of PrAO or MAO activities was based on an assay technique that uses Amplex Red/resorufin fluorescence, which has been successfully applied to the assay of other copper-containing amine oxidases [35]. However, numerous chemical species that quench resorufin-derived fluorescence blunt the signal and thereby falsely appear to inhibit enzyme activity, but do not interact directly with amine oxidases [32,35,36]. Thus, before testing the effects of natural N-methylated derivatives of xanthine such as theobromine, theophylline, caffeine, 7-methylxanthine, and paraxanthine (the chemical formula of these is available in Appendix A), it was necessary to check whether these compounds and others used in the assay interfered with the hydrogen peroxide-dependent fluorescent signal. This was performed in preliminary assays without any hScAT. None of the classical substrates and inhibitors used as references for determining PrAO or MAO activity generated interference with the fluorescent signal (Appendix A). Among the vehicles tested for methylxanthine solubilization, only 100 mM NaOH impaired the detection of 5 µM H_2_O_2_, while DMSO was without any notable effect up to 3700 mM (Appendix A). For the methylxanthines on their own, no impairment of hydrogen peroxide detection was detected with the highest dose studied, 2.5 mM, even for the non-natural methylxanthine derivative, IBMX. In contrast, the methylxanthine precursor, uric acid, greatly reduced the fluorescence readings at concentrations as low as 1.0 mM (Appendix A). Such behavior was unexpected, but similar to that previously found with various other natural compounds such as resveratrol or quercetin (Appendix A).

Since no notable interference was found between methylxanthines and the fluorescent signal generated by the chromogenic mixture in the presence of hydrogen peroxide, it was feasible to measure amine oxidase activity in hScAT preparations and to study its putative regulation by these compounds. However, methylxanthines are generally most active near their limit of water solubility [37], and require vehicles for their solubilization. It was therefore necessary to verify the influence of these solubilizing vehicles on amine oxidase activity.

### 3.2. Influence of Increasing Doses of Vehicles on Primary Amine Oxidase Activity in Human ScAT

The pure DMSO solution purchased from Sigma-Aldrich-Merck was estimated to be 14.8 M and was diluted with water to dissolve several methylxanthines. Since the methylxanthines solubilized in their vehicle represented a quarter of the volume distributed in each well of the black microplates (50/200 µL), the final DMSO concentrations present in the assays ranged from 3700 mM (for compounds dissolved in pure DMSO such as theobromine) down to 1850 mM (for xanthines dissolved in 50% v/v DMSO). This occurred when methylxanthines were tested at higher final concentrations of 2.5 mM. Typically, lower DMSO concentrations (<< 925 mM) were routinely used in the assays when solubilized xanthines were diluted for tests at millimolar or sub-millimolar doses. Figure 1 indicates that increasing doses of DMSO did not influence benzylamine oxidation by hScAT in the range of 148–1480 mM. However, when present at 2220–3700 mM, DMSO significantly impaired PrAO activity (Figure 1). Therefore, the apparent effects of the higher doses of several xanthines needed to be corrected when their vehicle was DMSO ≥ 2220 mM, while no correction was required for the lower concentrations of this organosulfur compound.

In spite of its recommended use for the dilution of uric acid [38], NaOH was not further used as a vehicle since it interfered with the fluorescence signal (Appendix A), and enhanced the signal attenuation induced by uric acid itself. Ethanol is recommended at 10% v/v for preparing IBMX stock solutions and did not exhibit similar inhibition, even when used as a vehicle up to a final level of 20% v/v (Figure 1). Finally, this verification step confirmed that benzylamine oxidation by hScAT is almost entirely due to SSAO/VAP-1 activity (i.e., to the human ortholog of bovine PrAO [12]), since 1 mM semicarbazide totally abolished the oxidation of 1 mM benzylamine (Figure 1).

### 3.3. Methylxanthine-Induced Inhibition of Benzylamine Oxidation in Human Adipose Tissue

All inhibition studies were carried out with 1.0 mM of benzylamine as the substrate since the Km of the PrAO activity found in hScAT was 126 µM (Appendix A). When present at micromolar doses, the methylxanthines tested did not inhibit the oxidation of 1 mM benzylamine by human adipose tissue. However, an inhibitory effect was observed with 1 mM of each tested xanthine, except paraxanthine (Figure 2). Caffeine and theobromine produced a remarkable 50% inhibition of PrAO activity when incubated at 2.5 mM with hScAT, when data were normalized to account for vehicle interference (Figure 2). In fact, the apparent inhibition seen with the higher dose of theobromine tested in the presence of 25% v/v DMSO inhibited more than 80% of benzylamine oxidation (not shown), but this effect was partly due to the vehicle. The calculated inhibition due to theobromine on its own exhibited the same magnitude of effect as that observed with the non-natural methylxanthine IBMX, dissolved in 10% v/v ethanol, or that of caffeine dissolved in water (i.e., two vehicles without any influence on PrAO activity).

The fact that inhibition was greatest with caffeine and theobromine indicates that the presence of N-methylation of the xanthine core at positions 3 and 7 is required for maximum inhibition. The same pattern was obtained with bovine PrAO [12]. It is possible that these compounds are interacting with one or more imidazole binding site(s) identified by previous studies [39]. Indeed, one of these sites is close to the active site and may overlap with the substrate binding site [40].

### 3.4. Methylxanthine-Induced Inhibition of Methylamine Oxidation by Human Adipose Tissue

To further characterize the capacity of methylxantines to inhibit a PrAO-catalyzed reaction, we determined their influence on the oxidation of another substrate: methylamine. This molecule is a physiologically relevant PrAO substrate present in mammals and comes either from the oxidation of adrenaline, or from exogenous sources, since it is naturally present in foods and beverages [41]. Since the Km values of methylamine for its oxidation by PrAO are in the 100–800 µM range [42], and in order to spare human material, a single test was performed with 0.5 mM methylamine as the substrate.

As expected, semicarbazide totally inhibited the oxidation of methylamine (99.8 ± 1.6% inhibition, not shown), which was resistant to pargyline inhibition (1.8 ± 5.7% inhibition, not shown), confirming that methylamine oxidation in hScAT is entirely due to PrAO, as recently reported [43]. The oxidation of methylamine was inhibited by 1 mM of either 7-methylxanthine, theobromine, caffeine, or IBMX (Figure 3). As with benzylamine, the inhibition was not complete, but significantly limited PrAO activity. The non-natural inhibitor IBMX appeared to be the strongest inhibitor tested, while caffeine and theobromine were the most efficient of the natural methylxanthines.

Inhibition of PrAO catalyzed methylamine oxidation followed a broadly similar pattern to that seen when benzylmine was a substrate with the exception that the non-natural methylxanthine, IBMX, was found to be highly effective, inhibiting activity by *ca*. 80%. This may be due to the lower bulk of this substrate, which may allow for greater access to an inhibitor binding site than benzylamine. Differences in patterns of inhibition between benzylamine and methylamine have been previously observed for the bovine enzyme [44].

### 3.5. Nature of the PrAO Inhibition Induced by Methylxanthines in Human Adipose Tissue

In an attempt to analyze the nature of the inhibition by methylxanthines on PrAO activity, the oxidation of increasing doses of benzylamine was explored at: 0.1, 0.2, 0.5, and 1.0 mM, with theobromine at 0, 0.1, 1.0, and 2.5 mM. A Lineweaver–Burk plot of these inhibition curves (Figure 4) illustrated the complexity of the pattern of inhibition and made determination of inhibition constants for theobromine difficult. It was noted at higher concentrations that these data showed some departure from Michaelis–Menten kinetics. This complex pattern indicated that the inhibition induced by theobromine was certainly not simply competitive.

While a Ki could not be determined, it was clear from these data that theobromine inhibited PrAO in a dose-dependent manner. The complex pattern of inhibition we observed may reflect issues with measurements made in tissue samples compared to experiments with purified enzymes. Nonetheless, it is possible to assess that theobromine at 2.5 mM inhibited half the PrAO activity in hScAT. The purified bovine PrAO had an IC_50_ of 0.27 mM for theobromine (as calculated using Ki = IC_50_ for a noncompetitive inhibitor, see [12]) Thus, the theobromine concentration required to achieve 50% inhibition for the human enzyme is higher than for the pure bovine enzyme. This may be due to inter-species differences and/or to tissue interference. It is possible that a significant proportion of the methylxanthine inhibitors may be retained in lipid droplets and not available for binding to enzymes in the hScAT homogenates. Such an explanation might also account for the complex kinetics observed. As with the bovine enzyme, it is possible that the methylxanthines in this study were binding to one or more imidazole binding site(s) identified in structural studies [40,45]. If blocking both of these binding sites caused inhibition of activity, then the complex kinetics we observed might be expected. Indeed, previous workers have noted the complexity of kinetics observed with this enzyme and proposed the presence of multiple binding sites on PrAO monomers [46].

### 3.6. Tyramine Oxidation by Human Adipose Tissue

Besides PrAO, human adipose cells also express MAO, an enzyme located at the outer mitochondrial membrane, mainly of the MAO-A form [29]. It was of interest to test whether methylxanthines interacted with adipose MAO, although the role of MAO in adipose tissues is less well understood than that of the brain MAOs, which are involved in neurotransmitter turnover, or the cardiac MAOs, considered as sources of reactive oxygen species during ischemia [47]. As well as exploring the inhibition of MAO found in human ScAT, this complementary investigation aimed to provide information about the selectivity of methylxanthines for inhibition of copper-containing PrAO versus FAD-containing MAO.

As with PrAO activity, verification of vehicle influence indicated that DMSO significantly impaired tyramine oxidation when present at ≥12.5 % v/v (≥1850 mM), in other words, the concentrations used as the vehicle for 2.5 mM paraxanthine, 7-methylxanthine, or theophylline (Figure 5). When used in pure form to dissolve and test theobromine at 2.5 mM, DMSO was not compatible with accurate determination of MAO activity. In addition, Figure 5 clearly indicates that in hScAT, the oxidation of 1 mM tyramine was resistant to semicarbazide, and inhibited by pargyline, an irreversible MAO inhibitor drug.

The Km for the oxidation of tyramine by hScAT homogenates was 66 µM (Appendix A) and the inhibition of MAO activity was examined using 1 mM tyramine as the substrate, with methylxanthines tested as inhibitors within the millimolar range only. Interestingly, IBMX was the most potent inhibitor and almost completely inhibited tyramine oxidation at 0.1 mM (Figure 6). Caffeine induced almost 75% inhibition at 2.5 mM. Paraxanthine, 7-methylxanthine, and theophylline only partially impaired tyramine oxidation. This was also the case for theobromine at 2.5 mM, as the presence of high amounts of DMSO did not allow for consistent measurements of MAO activity.

To circumvent the problem of methylxanthine solubility, especially in the case of theobromine, we tested for inhibition at a lower concentration of substrate and observed that the oxidation of 0.1 mM tyramine was significantly inhibited with 1 mM theobromine as well as by 7-methylxanthine, theophylline, or caffeine, irrespective of their different vehicles. Tyramine oxidation was not altered by semicarbazide or by lower doses of DMSO (Table 1).

At 0.1 mM tyramine, the reaction rate was equivalent to 63.3% of the rate at saturating tyramine. At this lower concentration, we clearly observed inhibition by all the methylxanthines tested. One mM caffeine diminished activity from 63.3% to 26.7% of the reference value. Taken together, all these data showed a clear impairment of tyramine oxidation by methylxanthines, but did not provide evidence of selectivity for PrAO vs. MAO inhibition.

## 4. Discussion

In the present study, the oxidation of benzylamine, methylamine, and tyramine by crude homogenates of adipose tissue obtained from overweight women confirmed that the oxidizing enzymes PrAO and MAO were abundantly expressed in this tissue and their activities were readily detectable without prior protein purification. The use of crude human biological material offered the advantage of determining the activity of the human native forms of PrAO and MAO, allowing for possible interplay with naturally occurring partners, regulators, or repressors to be examined. The clear-cut sensitivity to semicarbazide, which abolished benzylamine and methylamine oxidation (while having no effect on tyramine oxidation), clearly confirmed that the hydrogen peroxide released in the presence of these substrates was dependent on PrAO activity. In turn, tyramine oxidation was largely obliterated by pargyline, thereby showing it was predominantly MAO-dependent.

A known limitation when using such crude biological material arises due to the presence of large amounts of fat in adipose tissue that can influence enzyme accessibility for methylxanthines that are poorly soluble in aqueous solutions. Under *in vivo* conditions, the methylxanthines are less likely to be trapped in adipocyte lipid droplets and are, therefore, more available to interact with membrane enzymes such as PrAO. Thus, in the present case, the effect of methylxanthine concentration on enzyme activity will be more complex than for a purified enzyme preparation. The true concentration reaching the active site cannot be known with certainty.

Obviously, our *in vitro* studies are only indicative of what can occur upon ingestion of the dietary compounds or drugs, and our observations on the amine oxidase inhibitory properties of methylxanthines need to be confirmed by pharmacokinetics and *in vivo* studies before drawing definitive conclusions. In this regard, it is important to note that the PrAO activity found in human adipose tissue is approximately one thousand times higher than that found in the plasma of obese subjects [28]. It can therefore be supposed that methylxanthines might exert substantial effects directly on fat depots. However, it cannot be precluded that the inhibition observed for adipose PrAO may also occur with the plasma soluble form. Indeed, it is still not well established whether changes in plasma PrAO activity influence the onset of the cardiometabolic complications of diabetes and obesity or are simply consequences of the pathological processes. It has been reported that serum VAP-1/SSAO (PrAO) concentration is increased with hyperglycemia [48], and should be considered as an adaptive phenomenon influencing the incident diabetes. In any case, the potential inhibitory effect of methylxanthines on novel targets such as human PrAO would suggest that the present study should be considered preclinical.

Since we were aware of the limited bioavailability of the methylxanthines, we included in our approach a test of several vehicles that included as least two criteria: their capacity to readily solubilize these compounds at millimolar doses, but also their ability to produce minimal interference in the fluorescent assay used for quantification of sub-micromolar doses of hydrogen peroxide. The latter issue could have been avoided by using other more reliable detection methods to determine amine oxidase activity such as that using vanillic acid [36], though these assay methods are less sensitive than the fluorometric assay. This alternative assay appears feasible in light of the richness of amine oxidase activity in human adipose tissue. In fact, the choice of the fluorimetric method rendered necessary a verification of the quenching properties of the tested agents. Although this step does not appear compulsory, it allowed the detection of false positive (i.e., drug compounds that impair hydrogen peroxide detection rather than actually inhibiting its generation during substrate oxidation by amine oxidases). Such interference was observed for uric acid, which hampered resorufin-based detection of hydrogen peroxide. Several polyphenols including resveratrol and quercetin were tested as reference points in the present study and our work confirmed their previously described behavior [32,49]. Indeed, the real bottleneck of the fluorimetric method for the PrAO or MAO assay is that it is almost impossible to determine whether such quenching agents act as substrates/inhibitors or do not interact with the enzymes. For the methylxanthine precursor uric acid, our current study cannot make any conclusions about its inhibitory properties, while for resveratrol, complementary studies (using radiochemical assays for MAO and PrAO activities) have confirmed that it behaves as a MAO inhibitor [32]. It was not the fluorescence quenching on its own that complicated our observations, but rather the direct impairment of enzyme catalytic activity by the use of a large amount of vehicle. This was specifically the case for DMSO, which influenced MAO activity, as already reported [36], and to a lesser extend for PrAO activity, which was reduced by 50% in the presence of the organosulfur compound at 25% v/v.

Large concentrations of vehicles were required since the methylxanthines were inactive at micromolar doses (at which they are water soluble), whereas they were inhibitory between 1.0 and 2.5 mM. It must be specified that, lowering the substrates below their Km values would have allowed us to use less inhibitors and vehicles but would dramatically reduce the signal-to-noise ratio. In spite of the high apparent IC_50_ values of methylxanthines found in the present study, our approach has shown that the nature of the inhibition exhibited by theobromine is not simply competitive. This may indicate that methylxanthine binding sites exists not only in the active site of the enzyme, but also in other locations involved in allosteric regulation. This is consistent with the observations by Holt et al., who suggested multiple binding sites [39,46] Alternatively, the inhibition may involve binding to one form of the enzyme only. It has been reported for bovine PrAO that caffeine and theobromine inhibition was non-competitive [12,44]. It is important to note that the observations made with tissue homogenates are known to be more challenging than for a purified enzyme preparation, and that the differences we observed between hScAT and bovine enzyme may reflect interference from tissue components such as sequestration of inhibitor in lipid droplets.

Although initially confusing, such in vitro effects for high doses of methylxanthines are neither unexpected nor artifactual. A requirement of millimolar concentration of methylxanthines has already been observed for the inhibition of phosphodiesterases, the antagonism of adenosine receptors [22] or the inhibition of bovine PrAO [12,44]. Therefore, our observations have to be considered as a confirmatory extrapolation to humans of the recently observed inhibitory properties of theobromine on bovine PrAO [12]. A confirmation of these previous observations would have better deciphered the mechanisms of action if performed on purified recombinant human proteins, but would not have provided the same insights. Indeed, the methylxanthines are medicines widely used for decades, and the agents tested here belong to both the families of phytochemicals and drug compounds: strictly speaking, they do not require further preclinical assays. Their inhibitory action on enzymes involved in inflammatory processes and in oxidative stress presents a novel aspect when performed on adipose tissue, when one considers that this tissue often develops low-grade inflammation and insulin resistance in obesity and concomitant type 2 diabetes.

When we examined inhibition of MAO, we showed that several methylxanthines inhibited MAO in humans. It is important to note that, by comparison to resveratrol and related stilbenes, theobromine inhibits PrAO to the same extent as MAO, while the stilbenes are reported to inhibit MAO more than PrAO. Since the stilbenes exhibit IC_50_ values in the micromolar range, they have been successfully used to develop novel selective MAO inhibitors [50].

Obviously there are other plant-derived products that inhibit PrAO at much lower concentrations than theobromine such as the hydrolyzable tannin geraniin, which is reported to have an apparent inhibition constant, Ki, in the micromolar range [51]. Eugenol inhibits MAO-A and MAO-B activities with IC_50_ values ranging between 5 and 25 µM [52]. Nevertheless, the methylxanthines, since they are the most widely ingested phytochemicals, can be considered as factors influencing amine oxidase activity in consumers. The sum of most of the dietary methylxanthines circulating in blood will probably reach levels in the low to mid micromolar region [53]. It can be hypothesized that the combined action of ingested PrAO inhibitors may cause some attenuation of its catalytic activity and may thus mitigate inflammatory processes. It is worth noting that glucosamine, used to mitigate osteoarthritis, is a time-dependent inhibitor of PrAO when tested at the millimolar concentration range [54], and can be added to the list of agents exhibiting both anti-inflammatory and inhibitory PrAO properties. Imidazoline ligands represent another a class of agents that interact with PrAO on binding sites that are distinct from the catalytic site [46], indicating that many modulators of the enzyme activity are not solely acting at the catalytic site. Such imidazoline ligands (clonidine, cirazoline) are able to interact with PrAO and MAO [39], as observed here for the methylxanthines, which contain an imidazole ring.

Another important finding of the study was that caffeine almost totally abolished the capacity of hScAT to oxidize tyramine. Such inhibition by caffeine of the native MAO found in human adipose depots (mainly of the A form [29]) is in total agreement with previous observations made on recombinant human enzymes, which reported reversible and competitive inhibition with Ki values of 0.7 mM and 3.8 mM for MAO-A and MAO-B, respectively [30]. This inhibition needs to be further explored in terms of adipocyte biology, since it has already been reported that tyramine influences lipid deposition, under both in vitro and in vivo conditions [55]. Another observation linking adipose tissue physiology and PrAO activity is that methylamine and benzylamine activate glucose transport in human adipocytes in a PrAO-dependent manner [43] and that PrAO inhibitors have been proposed as anti-obesity agents [18,56].

Finally, two aspects need to be discussed in view of the observations reported by our preclinical approach. The first one deals with phytotherapy, traditional medicine, or personalized nutrition. Since methylxanthines coexist in various medicinal plants, their individual moderate inhibitory properties can be cumulative in consumers, leading to a relevant and sustained PrAO/MAO inhibition upon ingestion of coffee, chocolates, or food supplements. Indeed, one has to not only take into account the plasma concentrations of each methylxanthine (e.g., varying from 1 to 10 µM for caffeine), but the sum of these and their metabolites achieved in normal or excessive consumers or in subjects with altered PrAO or MAO expression. The notion of a “matrix effect” and synergism should be taken into consideration when further analyzing the beneficial and adverse effects of methyxanthine-rich foods, beverages, or supplements. The second aspect deals with the use of methylxanthines as a scaffold for the design and screening of novel inhibitors of either PrAO or MAO of the A and B forms [12,30]. Even if the inhibitions evidenced *in vitro* in this study are not relevant for nutritional or pharmacological trials in humans, in view of the high concentrations required, structural modifications of methylxanthines can likely lead to more potent inhibitors of MAO-A and MAO-B or PrAO. Thus, for example, targeting the imidazole binding site on PrAO using xanthine analogs is a worthwhile avenue of research indicated by this work. The increasing number of patents related to the anti-inflammatory properties of agents blocking PrAO activity, and thereby limiting the lymphocyte extravasation at the sites of inflammation, leads us to suppose that methylxanthine-derivatives may have the required properties to improve this novel therapeutic approach.

Meanwhile, the PrAO and MAO inhibition provoked by methylxanthines should be considered in ongoing and future investigations using these drugs for in vitro experiments where high concentrations are often used to block phosphodiesterases or adenosine receptors such as, for instance, the inclusion of IBMX in differentiation media for preadipocyte cell lines or in prolonged treatments of rodent [57] or human [58] adipocytes.

The levels of methylxanthines attained in vivo (bioavailabilty) have been the subject of research and there is evidence of some variation by gender and by age [59]. However, some studies do report levels as high as 63µm (for theobromine following chocolate consumption) in plasma see [53]. Of course, higher levels may be achieved in individuals who, for example, consume significant amounts of caffeine-containing energy drinks. If the Ki for the human PrAO toward theobromine, for example, were assumed to be similar to that of the bovine enzyme, we might expect 63 µm theobromine to reduce its activity by around 20%. While this is clearly significant, it remains speculative until detailed studies with pure enzyme are carried out. Nonetheless, it is known that these compounds have a wide variety of health benefits when present in the diet, especially on neurodegenerative disease [4,5,60,61] and must, therefore, reach levels sufficient to exert pharmacological action. The true levels of methylxanthine interacting with the enzyme under the conditions of this study were impossible to estimate due to the possibility of interaction with lipids and tissue components. This can only be determined by more detailed *in vivo* studies with human volunteers. The only *in vivo* study reported to date was carried out in rats [62]. That study reported that “caffeine was found to be present in all tissues after administration for 10 days and accumulated for 25 days” and that rising caffeine levels correlated with loss of PrAO activity. Thus, it has been established in rat that ingestion of a methylxanthine can lead to PrAO inhibition and that such inhibition can accumulate over time.

## 5. Conclusions

This study confirms that the bulk of benzylamine and methylamine oxidizing activity in hScAT is carried out by PrAO, and the bulk of tyramine oxidation is carried out by MAO. We have also shown, for the first time, clear evidence that human adipose PrAO activity is strongly inhibited by some methylxanthines (theobromine and caffeine) and not others, albeit at high concentrations. Among the naturally occurring methylxanthines, methylation at positions 3 and 7 of the xanthine nucleus appears to be important for inhibition, identical to the pattern observed for the bovine enzyme [12]. Surprisingly, the non-natural methylxanthine, IBMX, which lacks this motif, was a strong inhibitor of both PrAO and MAO.

It was not possible to clearly deduce the effect of dietary levels of theobromine and caffeine on PrAO from this work due to the difficulty in measuring a Ki in crude hScAT homogenates. However, it is clear that when administered as drugs, the resulting amounts of some methylxanthines could inhibit this enzyme. The only *in vivo* study reported to date showed that caffeine, when administered to rats, inhibited PrAO in adipose tissues [62]. Our study clearly supports a similar observation for humans.

It is also of considerable interest that roughly 75% of the tyramine oxidation by hScAT was inhibited by pargyline. The non-natural compound IBMX inhibited this activity by an extraordinary 80%. This was followed by caffeine, which inhibited tyramine oxidation by around 75%. These findings show that it is important to consider inhibition of MAO when examining the effect of therapeutic doses of some methylxanthines and raises the intriguing possibility that a xanthine scaffold could be the basis for design of novel PrAO and MAO inhibitors.

In summary, the clinical significance of this work is that it will provide the first baseline data relating the methylxanthine structure to inhibition efficacy for PrAO and MAO in hScAT. It will act as a stimulus to further investigate the combined action of methylxanthines on human PrAO and MAO. It will also prompt an exploration of methylxanthine derivatives as possible therapeutics for a wide range of disorders. Finally, it suggests that the large number of existing xanthine derivatives currently in use as therapeutics should be tested for their effect on PrAO and MAO.

## Figures and Tables

**Figure 1 medicines-07-00018-f001:**
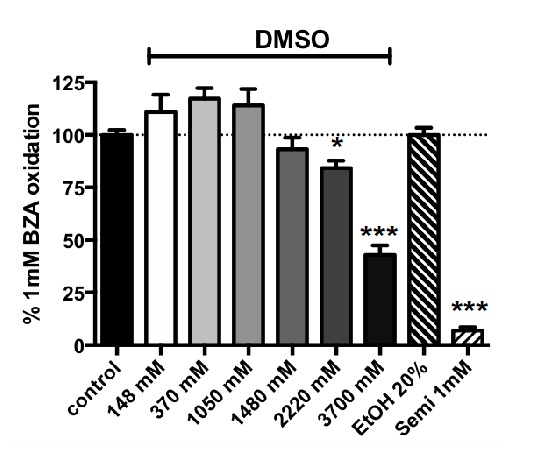
Influence of DMSO, ethanol, and semicarbazide on 1 mM benzylamine oxidation by human adipose tissue homogenates. Assays were performed at 37 °C as described in Materials and Methods with hSCAT homogenates and 1 mM benzylamine as the control and the indicated final concentration of: DMSO: dimethyl sulfoxide; EtOH: ethanol; Semi: semicarbazide. The highest tested dose of DMSO (3700 mM) corresponds to its final concentration when used in pure form as a vehicle to solubilize a xanthine at 10 mM, then testing it at 2.5 mM final. The amount of H_2_O_2_ generated during 30-min oxidation of benzylamine (1.40 ± 0.20 nmoles/mg protein/min) was set at 100% (dotted line), while baseline with only hScAT was set at 0 %. Mean ± SEM of n = 7–27 different individual preparations. Different from control at: * *p* < 0.05; *** *p* < 0.001 by one-way ANOVA test.

**Figure 2 medicines-07-00018-f002:**
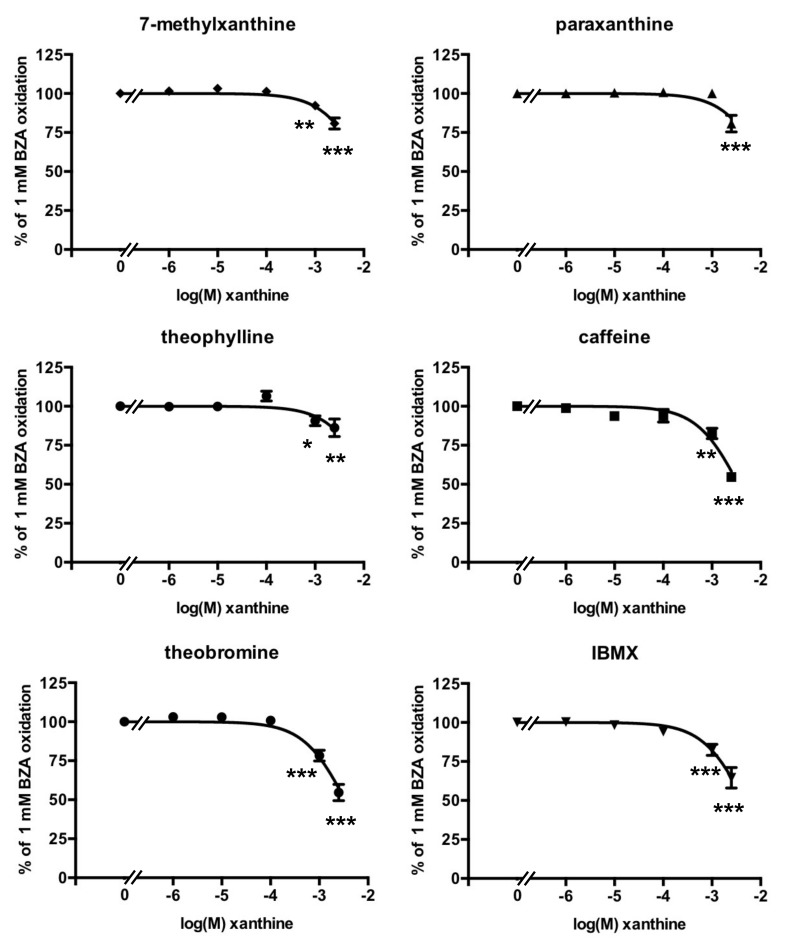
Dose-dependent inhibition of benzylamine oxidation by methylxanthines. The control consisted of a standard assay of PrAO-dependent oxidation of 1 mM benzylamine (BZA), performed in triplicate for each of the 27 individual tissue samples, and set at 100%. Mean ± SEM of the number of observations indicated for each agent, added over a concentration range from 1 µM to 2.5 mM: 7-methylxanthine: 8–20; Paraxanthine: 8–10; Theophylline: 8–20; Caffeine: 8–24; Theobromine: 8–20; IBMX: 8–14. The differences from control (plotted as 0 on the X-axis) at: * *p* < 0.05; ** *p* < 0.01; *** *p* < 0.001 were analyzed by one-way ANOVA. In most cases, the SEM bars lie within the symbol.

**Figure 3 medicines-07-00018-f003:**
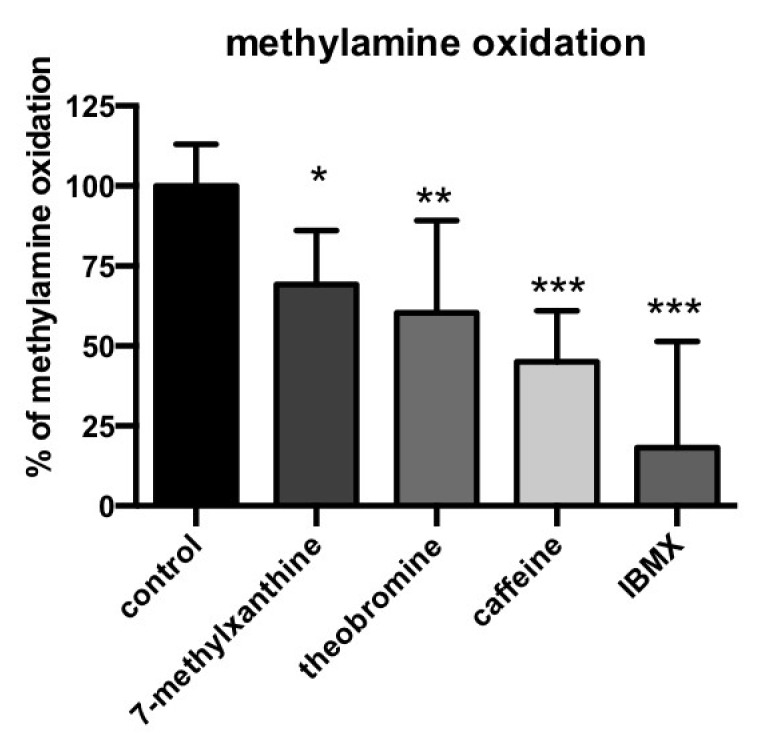
Inhibition by methylxanthines of the methylamine oxidation by human PrAO activity. The control condition consisted of the oxidation of 0.5 mM methylamine by hScAT homogenates during 30 min at 37 °C and was set at 100% with the baseline of hydrogen peroxide release set at 0%. Each methylxanthine was added at 1 mM final and preincubated for 10 min before the addition of methylamine. Each column is the mean + SEM of seven individuals, each performed in duplicate. Different from control (dark column) at: * *p* < 0.05; ** *p* < 0.01; *** *p* < 0.001 by one-way ANOVA.

**Figure 4 medicines-07-00018-f004:**
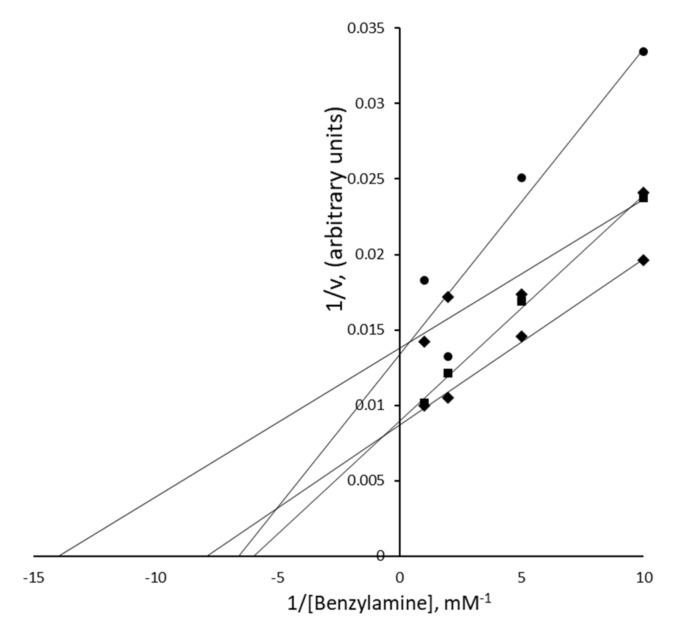
Inhibition of hScAT PrAO by theobromine. The concentration of benzylamine varied between 0.1 and 1.0 mM in the presence of increasing concentrations of theobromine: 0, 0.1, 1.0 and 2.5 mM. The concentration of theobromine used were: 0 mM (diamonds), 0.1 mM (squares), 1.0 mM (diamonds), and 2.5 mM (circles).

**Figure 5 medicines-07-00018-f005:**
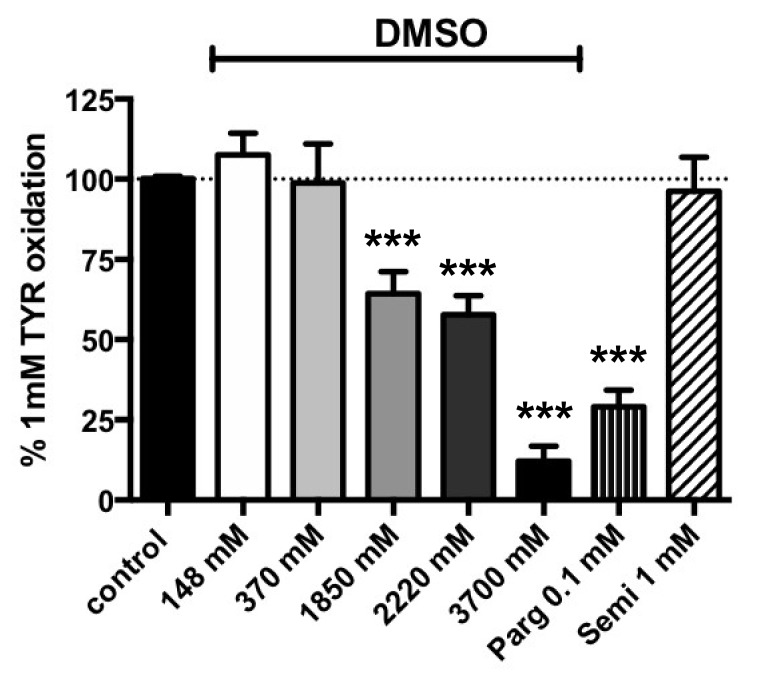
Interaction between DMSO vehicle and tyramine oxidation by human abdominal ScAT. Assays were performed at 37 °C with homogenates and 1.0 mM tyramine as a control plus the indicated final concentration of: DMSO, Parg: pargyline; Semi: semicarbazide. The highest tested dose of DMSO (3700 mM) corresponds to its final concentration (25% v/v) when used as the vehicle for solubilization of 2.5 mM theobromine. The amount of H_2_O_2_ generated during 30-min oxidation of tyramine (averaging 0.42 nmoles/mg protein/min) was set at 100% (dotted line), with the baseline set at 0%. Mean ± SEM of n = 8–12 different individual preparations. Different from control at: *** *p* < 0.001, determined by one-way ANOVA.

**Figure 6 medicines-07-00018-f006:**
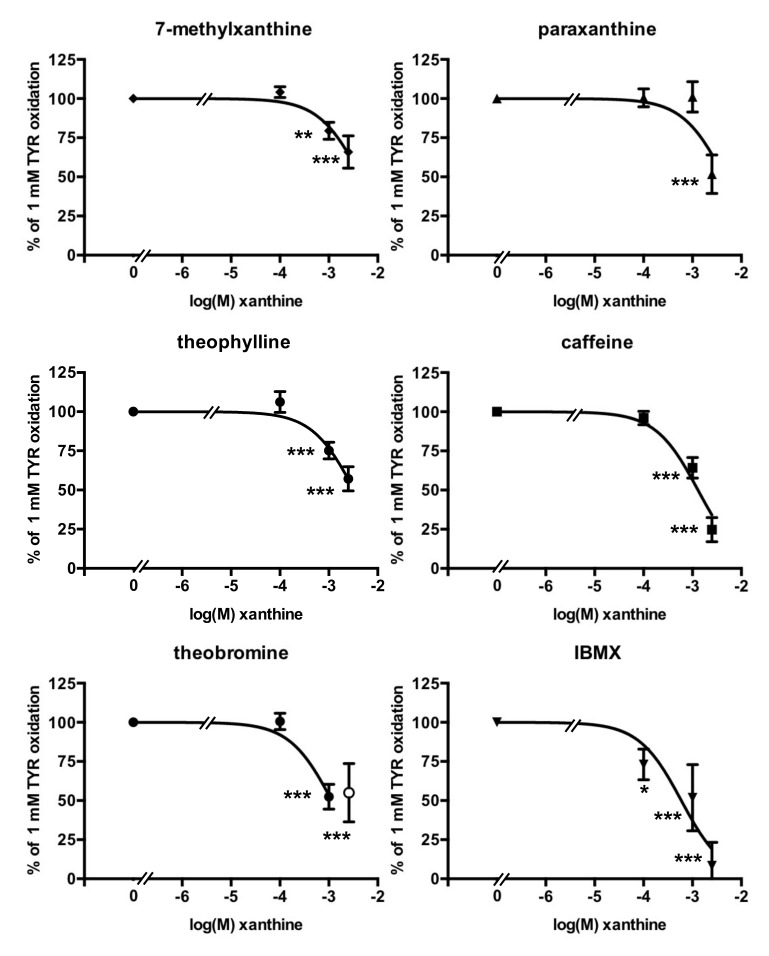
Dose-dependent inhibition of tyramine oxidation by methylxanthines. The control consisted of a standard assay of MAO-dependent oxidation of 1.0 mM tyramine (TYR) by hScAT homogenates, performed in triplicate for each of the 25 individuals and set at 100%. Mean ± SEM of the number of observations indicated for each agent: 7-methylxanthine: 14–16; Paraxanthine: 8; Theophylline: 20; Caffeine: 8–18; Theobromine: 8–19; IBMX: 7. For theobromine, the (open) symbol at 2.5 mM is different from those of lower doses to denote the extensive influence of 25% v/v DMSO vehicle needed for this xanthine at this dose, which could not be accurately estimated. The difference from control (plotted as 0 on the X-axis) at: * *p* < 0.05; ** *p* < 0.01; *** *p* < 0.001 was determined by one-way ANOVA.

**Table 1 medicines-07-00018-t001:** Inhibition of the oxidation of 0.1 mM tyramine in human adipose tissue.

Condition	Percentage of Tyramine Oxidation
Tyramine 1 mM	100% reference
Tyramine 0.2 mM	76.9 ± 4.6 (9) NS
Tyramine 0.1 mM	63.3 ± 6.1 (17)
Tyr 0.1 mM + semicarbazide 1 mM	65.2 ± 7.8 (4) NS
Tyr 0.1 mM + DMSO 148 mM	52.2 ± 5.8 (10) NS
Tyr 0.1 mM + theophylline 1 mM	28.8 ± 5.0 (15) ***
Tyr 0.1 mM + theobromine 1 mM	27.6 ± 8.2 (11) ***
Tyr 0.1 mM + caffeine 1 mM	26.7 ± 8.3 (9) ***
Tyr 0.1 mM + 7-methylxanthine 1 mM	25.1 ± 2.8 (10) ***

Values are expressed as a percentage of the oxidation of 1 mM tyramine by human adipose tissue preparations. Mean ± SEM of the number of individuals indicated in parentheses. NS: not significantly different from Tyr 0.1 mM alone; ***: different from Tyr 0.1 mM at *p* < 0.001.

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
