# Peer review of "Methylxanthines Inhibit Primary Amine Oxidase and Monoamine Oxidase Activities of Human Adipose Tissue"

_medicines, 2020, doi:10.3390/medicines7040018_

Round 1

Reviewer 1 Report

In the manuscript entitled “Methylxanthines Inhibit Primary Amine Oxidase and Monoamine Oxidase Activities of Human Adipose Tissue,” the authors reported that methylxanthine compounds inhibit PrAO and MAO activities. I read this manuscript with carefully, and found some doubts as following. Therefore, I concluded that the authors should re-constrict the strategy.

  • In this study, many kinds of solvent conditions were employed, because of the solubility of methylxanthine compounds. For example, theobromine was dissolved in pure DMSO. Basically, we know that theobromine, it is quite difficult to solve in water. So, the authors think about theobromine after absorbed into body, it cannot be circulating in the blood (water base)? If so, I cannot agree with it. Small amounts of theobromine can circulate in the blood stream, and exert beneficial effects. In my opinion, concentrations employed in this study were quite high, which is impossible for their bioavailability.
  • In this study, abdominal ScAT samples were collected from 27 overweight women. Therefore, individual difference about 27 ScAT samples should be mentioned at first. And then, what is the difference between overweight and normal (BMI < 25), and also between women and men should be mentioned.
  • I could not find the data, which can use Student’s t-test in this manuscript.

Author Response

Please find our response in the attachment.

Reviewer 2 Report

 The authors describe the anti-inflammatory effects of methylxanthines, indicating that the action is due in addition to the inhibition of phosphodiesterase and the adenosine receptor antagonism, also to the inhibition of PrAO and MAO 

I suggest that authors add thess recent manuscripts to the introduction

 Badshah H, Ikram M, Ali W, Ahmad S, Hahm JR, Kim MO. Caffeine May AbrogateLPS-Induced Oxidative Stress and Neuroinflammation by Regulating Nrf2/TLR4 inAdult Mouse Brains. Biomolecules. 2019 Nov 8;9(11). pii: E719.   Janitschke D, Nelke C, Lauer AA, Regner L, Winkler J, Thiel A, Grimm HS,Hartmann T, Grimm MOW. Effect of Caffeine and Other Methylxanthines onAβ-Homeostasis in SH-SY5Y Cells. Biomolecules. 2019 Nov 2;9(11).

Petrucci R, Chiarotto I, Mattiello L, Passeri D, Rossi M, Zollo G, Feroci AM. Graphene Oxide: A Smart (Starting) Material for Natural MethylxanthinesAdsorption and Detection. Molecules. 2019 Nov 21;24(23). pii: E4247.   

Barišić V, Kopjar M, Jozinović A, Flanjak I, Ačkar Đ, Miličević B, Šubarić D, Jokić S, Babić J. The Chemistry behind Chocolate Production. Molecules. 2019 Aug 30;24(17). pii: E3163.  

Mayorga-Gross AL, Esquivel P. Impact of Cocoa Products Intake on Plasma andUrine Metabolites: A Review of Targeted and Non-Targeted Studies in Humans.Nutrients. 2019 May 24;11(5). pii: E1163.   

Janitschke D, Nelke C, Lauer AA, Regner L, Winkler J, Thiel A, Grimm HS,Hartmann T, Grimm MOW. Effect of Caffeine and Other Methylxanthines onAβ-Homeostasis in SH-SY5Y Cells. Biomolecules. 2019 Nov 2;9(11).  

Santos CIAV, Ribeiro ACF, Esteso MA. Drug Delivery Systems: Study of InclusionComplex Formation between Methylxanthines and Cyclodextrins and TheirThermodynamic and Transport Properties. Biomolecules. 2019 May 20;9(5) 

Rojo-Poveda O, Barbosa-Pereira L, Mateus-Reguengo L, Bertolino M, Stévigny C, Zeppa AG. Effects of Particle Size and Extraction Methods on Cocoa Bean ShellFunctional Beverage. Nutrients. 2019 Apr 17;11(4). pii: E867. 

The concentration used were cytotoxic ?, this is important for drug safety

In the conclusions , the Authors should highlight the possible clinical significance of their findings

Author Response

(The authors gave the same response as above.)

Round 2

Reviewer 1 Report

I checked the revised manuscript with the authors comments, and think that the paper is well-written with reasonable conclusions, the authors’ approach is clear and simple to follow, and interesting data have been achieved with this study.